# First-Line Combination Treatment with Low-Dose Bipolar Drugs for ABCB1-Overexpressing Drug-Resistant Cancer Populations

**DOI:** 10.3390/ijms24098389

**Published:** 2023-05-07

**Authors:** Sungpil Yoon, Hyung Sik Kim

**Affiliations:** School of Pharmacy, Sungkyunkwan University, 2066 Seobu-ro, Jangan-gu, Suwon 16419, Republic of Korea

**Keywords:** bipolar drugs, heterogeneous resistant cancer population, first-line combination treatment, ABCB1 (or P-gp) overexpression

## Abstract

Tumors include a heterogeneous population, of which a small proportion includes drug-resistant cancer (stem) cells. In drug-sensitive cancer populations, first-line chemotherapy reduces tumor volume via apoptosis. However, it stimulates drug-resistant cancer populations and finally results in tumor recurrence. Recurrent tumors are unresponsive to chemotherapeutic drugs and are primarily drug-resistant cancers. Therefore, increased apoptosis in drug-resistant cancer cells in heterogeneous populations is important in first-line chemotherapeutic treatments. The overexpression of ABCB1 (or P-gp) on cell membranes is an important characteristic of drug-resistant cancer cells; therefore, first-line combination treatments with P-gp inhibitors could delay tumor recurrence. Low doses of bipolar drugs showed P-gp inhibitory activity, and their use as a combined therapy sensitized drug-resistant cancer cells. FDA-approved bipolar drugs have been used in clinics for a long period of time, and their toxicities are well reported. They can be easily applied as first-line combination treatments for targeting resistant cancer populations. To apply bipolar drugs faster in first-line combination treatments, knowledge of their complete information is crucial. This review discusses the use of low-dose bipolar drugs in sensitizing ABCB1-overexpressing, drug-resistant cancers. We believe that this review will contribute to facilitating first-line combination treatments with low-dose bipolar drugs for targeting drug-resistant cancer populations. In addition, our findings may aid further investigations into targeting drug-resistant cancer populations with low-dose bipolar drugs.

## 1. Introduction

Tumors associated with poor survival comprise high-grade, metastatic, advanced, and aggressive cancers because they are difficult to remove with surgery and can recur in a short time or eventually with the use of currently available chemotherapeutic drugs. These tumors include ovarian cancer, pancreatic cancer, lung cancer, glioma, and triple-negative breast cancer (TNBC). Therefore, other treatment strategies are needed to remove the tumor or delay tumor recurrence. Tumors comprise a heterogeneous population that mainly contains chemotherapeutic drug-sensitive cancer cells. However, a small proportion of tumors are composed of multidrug-resistance (MDR) cancer cells, which include cancer stem cells (CSCs) [1,2].

A critical mechanism for evading chemotherapeutic toxicity is the overexpression of ABCB1 protein or P-glycoprotein (P-gp) on their membranes and the active pumping of chemotherapeutic drugs as substrates from cancer cells to reduce their intracellular toxicity [1,2,3,4]. Although existing chemotherapeutic treatments for tumors reduce proliferation and apoptosis in drug-sensitive cancer populations, they stimulate an increase in drug-resistant cancer populations. In addition, drug-induced resistant cancer cell populations continue to rise, resulting in tumor recurrence. Additionally, tumor recurrence has led to an increase in the number of resistant cancer cells. However, chemotherapeutic drug treatments have not demonstrated effectiveness in sensitizing these tumors and reducing their size or proliferation. Resistant cancer populations show increased P-gp overexpression in their membranes [5,6,7]. Increasing P-gp overexpression has been recently identified as an important mechanism for overcoming chemotherapeutic drug resistance in cancer cells in heterogeneous populations [2,3]. They increase the gene copy numbers of ABCB1 via aneuploidy or uneven chromosomal segregation, resulting in the growth of drug-resistant tumors and recurrence after treatment with chemotherapeutic drugs.

Targeting the P-gp-overexpressing drug-resistant cancer cell population could delay tumor recurrence. Therefore, it is important to reduce drug-resistant cancer cell populations through apoptosis in heterogeneous tumor populations during initial or first-line chemotherapeutic drug treatment. Considering that these drug-resistant cancer cells show an increased overexpression of ABCB1 on their membranes, it is assumed that combination treatment with P-gp inhibitors can increase the possibility of successful treatments to shrink tumor size over a relatively long time and delay tumor relapse (Figure 1).

The selection of P-gp inhibitors is important in the application of first-line combination treatment as a protocol for tumor treatment. Notably, P-gp inhibitors cause side effects in patients with cancer [8,9,10]. In addition, P-gp inhibitors have shown toxicity in normal tissues, and their use as a treatment approach has been shown to hinder the immune cell attack on tumors in cancer microenvironments [9,10]. Therefore, reducing the toxicity of P-gp inhibitors is necessary. For faster clinical trials, researchers may use drug repositioning with FDA (United States Food and Drug Administration)-approved drugs that have a long history of clinical application and are easily used to target resistant cancer cells once they are identified to inhibit P-gp activity or target P-gp-overexpressing drug-resistant cancer cells [11,12,13].

Orally administered bipolar drugs have been shown to target antagonists for dopamine receptors on brain cell membranes [14,15]. Bipolar drugs have shown anticancer activity in various models for a long time [16,17,18,19,20], with clinical trials being held in cancer patients. Bipolar drugs for glioma treatment are beneficial, and their ability to penetrate the blood-brain barrier at low doses has been tested. The failure of chemotherapeutic drugs in the treatment of gliomas results from their limited penetration of the blood-brain barrier [14,15]. Bipolar drugs sensitize various types of drug-resistant cancer cells, while few of them possess strong P-gp inhibitory activity and increase sensitization in anticancer drugs for the combined therapy of P-gp-overexpressing resistant cancer cells [21,22,23,24,25]. Since combined therapy with low-dose bipolar drugs can increase the cytotoxicity of drug-resistant cancer cells, they can be used as a first-line combination treatment for heterogeneous drug-resistant cancer populations. Various bipolar drugs with different degrees of P-gp inhibitory activity have been identified [22,23,24,25]. Few low-dose bipolar drugs can sensitize P-gp-overexpressing drug-resistant cancer cells with weak P-gp-inhibitory activity, whereas others can sensitize cells with strong P-gp-inhibitory activity. Several combinations of bipolar drugs can contribute to personalized medicine by efficiently targeting drug-resistant cancers. Due to the fact that bipolar drugs are not substrates for P-gp efflux [22,23], they can efficiently target and increase the cytotoxicity of P-gp-overexpressing drug-resistant cancer cells.

This review describes several bipolar drugs that have recently been shown to sensitize P-gp-overexpressing drug-resistant cancer cells. These drugs can be used as potential candidates for first-line combination treatments and repositioned at low doses. The drugs include aripiprazole, fluphenazine, thioridazine, trifluopherazine, pimozide, quetiapine, loxapine, and iloperidone [14,15,16,17,18,19,20] (Table 1). Combination therapy with low-dose bipolar drugs is the focus of the discussion in the present review. Therefore, herein, we discuss combination therapies with low-dose bipolar drugs. This review can provide additional combination treatment options for these bipolar drugs as first-line combination treatments (Figure 1). In addition, we believe that this review will facilitate further clinical trials of P-gp-overexpressing drug-resistant cancer cells targeting drugs to kill small portions of CSCs or drug-resistant cancer populations.

## 2. Candidate Bipolar Drugs for First-Line Combination Treatments

### 2.1. Aripiprazole

Although the sensitization effects of aripiprazole as a single therapy for drug-resistant cancer cells have not been reported, a recent study demonstrated the anticancer activity of aripiprazole as a single treatment [26]. This study showed that aripiprazole increased apoptosis in various malignant cancer cell lines. The sensitization mechanism involves the downregulation of MMP9, phosphorylated PI3K, and pStat3. Most importantly, this study demonstrated that aripiprazole mainly targets the reduction in Src kinase activity to sensitize malignant cancer cells with the help of an in vivo xenograft mouse model. Aripiprazole treatment significantly reduced the injected colon cancer cell mass in mice [26]. In addition, the xenograft demonstrated that the amount of aripiprazole did not harm healthy tissues because the mouse body weight was unaffected in the aripiprazole-treated group. Another study using a single treatment of aripiprazole showed that it selectively increased apoptosis in breast cancer cell lines based on the screening of 97 chemical drugs [27], suggesting that aripiprazole could be a targeted repositioned drug for the treatment of breast cancer. Both of these studies involved single treatments that required a high dose of aripiprazole for sensitizing cancer cells; low-dose aripiprazole can be used to sensitize cancer cells in a combination treatment.

Recent studies demonstrated that low-dose aripiprazole is a sensitizer of combination treatments in in vitro cancer cell lines [21,23,28]. With the administration of aripiprazole at a low-dose of ˂2.5 μM, cancer cells have phenotypically normal proliferation and may grow similarly to untreated cancer cells. In addition, ˂2.5 μM aripiprazole has no apoptotic effects on in vitro cancer cell lines [21,23,28]. However, combination treatment with low-dose aripiprazole increased apoptosis in radiation-treated head and neck cancer cell models [28]. Furthermore, this study showed that the sensitization mechanism of radiation + aripiprazole combined therapy mainly involved an increase in reactive oxygen species. The authors concluded that aripiprazole could be a potential radiosensitizer candidate. In addition, the study demonstrated the combined effects in a mouse xenograft model, supporting the idea that aripiprazole combination treatment can be applied faster in clinical trials.

Bipolar drugs have structures similar to those of dopamine [14]. Considering that dopamine receptors are present on cellular membranes, aripiprazole may also target P-gp on the cell membranes. Some bipolar drugs have been shown to inhibit P-gp expression in in vitro cancer cell models [22,24,25]. Recent studies have demonstrated that low-dose aripiprazole has strong P-gp inhibitory activity [21,23] (Figure 2A–C). The study showed that 2.5 μM aripiprazole could adequately inhibit the efflux ability of rhodamine 123 substrates in P-gp-overexpressing drug-resistant cancer cells compared to 10 μM verapamil or 5 μM reserpine (the positive controls). In this analysis, the P-gp-inhibitory activity of aripiprazole was high after 4 h (a short time) of treatment, suggesting that aripiprazole can directly inhibit P-gp via docking contact and not by transcriptional regulation of P-gp. Aripiprazole treatment for 24 h inhibited P-gp activity, similar to the effect of 4 h treatment, indicating that low-dose aripiprazole can inhibit P-gp activity over a long interval.

Most importantly, combined therapy with low-dose aripiprazole increased the cytotoxicity of P-gp-overexpressing drug-resistant cancer cells against antimitotic drugs. This indicates that the strong P-gp-inhibitory activity of low-dose aripiprazole prevents the efflux of antimitotic drugs in P-gp-overexpressing immune cancer cells. Based on annexin V staining, C-PARP levels, and cell cycle analysis, combination treatments with aripiprazole increased apoptosis and G2 arrest in antimitotic drug-treated P-gp-overexpressing resistant cancer cells. In addition, the molecular mechanisms of this combination treatment resulted in increased pRb and pH2AX levels, suggesting that DNA damage increased G2 arrest and apoptosis. The critical growth signaling pathway, the ERK pathway, was highly downregulated by aripiprazole combination treatment for P-gp-overexpressing resistant cancer cells treated with antimitotic drugs. Low-dose aripiprazole had no effect on either drug-sensitive or drug-resistant cancer cells, suggesting that it can be clinically useful to minimize normal toxicity. Even 1 μM aripiprazole had a P-gp inhibitory activity similar to that of 10 μM verapamil, implying that reduced cellular toxicity can be accomplished by targeting P-gp-overexpressing resistant cancer cells with a combination treatment of low-dose aripiprazole. In addition, it is noteworthy that aripiprazole is not a substrate for P-gp efflux because the IC50 is similar between drug-sensitive and drug-resistant cancer cells. Aripiprazole could be combined with various chemotherapeutic drugs, including vincristine, eribulin, and vinorelbine. This suggests that low-dose aripiprazole can be combined with any chemotherapeutic drug as a first-line combination partner. Among these anti-mitotic drugs, eribulin has recently been developed for metastatic, drug-resistant, and advanced cancer types, such as TNBC [29,30]. Considering that eribulin can target advanced and drug-resistant cancer cells, combined therapy with aripiprazole and eribulin may be a promising first-line combination chemotherapy for P-gp-overexpressing resistant cancer populations.

Moreover, a recent study demonstrated that aripiprazole had the lowest dose among 15 bipolar drugs for sensitizing antimitotic drugs-treated P-gp overexpressing resistant cancer cells [21,23]. Considering that low-dose aripiprazole has the strongest P-gp inhibitory activity among the 15 bipolar drugs compared (Table 1 and Figure 2A–C), the highest sensitization effect of aripiprazole resulted from its strong P-gp inhibitory activity in antimitotic drug-treated P-gp-overexpressing resistant cancer cells. Since reduced toxicities are proportional to low-dose drugs, aripiprazole is the most promising bipolar drug for a first-line combination treatment for resistant cancer populations. Considering that the strong P-gp inhibitory activity of low-dose aripiprazole has recently been demonstrated, intensive studies in both in vitro and in vivo animal models should be conducted for faster clinical trials.

### 2.2. Fluphenazine

Fluphenazine is designated as an essential drug for managing bipolar diseases [31] and can be administered to cancer patients as a repositioning drug once its novel anticancer mechanisms are established. Different from aripiprazole, fluphenazine has been intensively studied in various cancer models, including brain, liver, oral, ovarian, lung, colon, breast, and leukemia models. These studies showed the cytotoxicity of a single treatment with fluphenazine, which has relatively high doses or concentrations of ˃10 μM. Cancer sensitization mechanisms by single fluphenazine treatment involve apoptosis via DNA fragmentation, reactive oxygen species, or Akt and Wnt signaling pathways [31]. Fluphenazine treatment also reduced the migration and invasiveness of metastatic cancer cells. In in vivo mouse xenograft models, fluphenazine was effective against poor survival-associated tumors such as TNBC, glioma, and ovarian cancer. Significantly, these treatments could reduce metastasis, which is the leading cause of poor survival rates in these cancers. Phase I and II clinical trials of fluphenazine in patients with leukemia are in progress. Furthermore, studies have shown that combination treatments with fluphenazine, such as in combination with paclitaxel, significantly induce apoptosis in breast cancer cell models [32]. However, few studies have reported sensitization of resistant or MDR cancers by co-treatment with fluphenazine.

Combination treatment with low-dose or 5 μM fluphenazine increased the cytotoxicity of eribulin-treated KBV20C oral cancer cells [23,25]. In contrast, a single treatment with low-dose fluphenazine or eribulin had no phenotypic effect on drug-resistant cancer cells. The mechanism of eribulin + fluphenazine sensitization involves increased DNA damage, G2 arrest, and apoptosis.

Single fluphenazine treatment has a similar IC50 in sensitive and P-gp-overexpressing resistant cancer cells [25], suggesting that fluphenazine is not a substrate for the efflux of P-gp. In this study, fluphenazine showed P-gp-inhibitory activity in an efflux pump-out assay using both calcein-Am and rhodamine 123 substrates in P-gp-overexpressing resistant KBV20C cells [25]. The study used P-gp-overexpressing resistant KBV20C cancer cells, which showed high eribulin resistance and had more than a 500-fold increase in IC50 compared to sensitive KB cells. Considering that eribulin is a recently developed antimitotic drug for metastatic cancers [29,30], low-dose fluphenazine can be regarded as a promising bipolar drug for first-line combination treatment of ABCB1-overexpressing heterogeneous populations. Fluphenazine analogs have shown P-gp inhibitory activity in in vitro human lymphocyte culture models [33]. Although fluphenazine has a lower P-gp-inhibitory activity and less sensitization effect than aripiprazole [23,24,25], low-dose fluphenazine exerts adequate sensitization effects on P-gp-overexpressing resistant cancer cells. Among the 15 bipolar drugs tested, low-dose fluphenazine was one of the top five drugs for sensitizing P-gp-overexpressing resistant cancer cells [21,23]. Therefore, fluphenazine may be considered part of a first-line combination treatment for resistant cancers in heterogeneous tumor populations. As personalized medicine becomes more important, either fluphenazine or aripiprazole combination treatment can be selected based on individual differences or tumor-organ type specificity.

### 2.3. Thioridazine

Thioridazine is one of the most widely studied repositioned bipolar drugs with anticancer activity [14,15,34]. The anticancer activity of thioridazine is attributed to its various cellular functions. The drug has been shown to increase apoptosis in various cancer models. Particularly, it induced apoptosis in leukemic cells without affecting normal cells [35]. Thioridazine also increased apoptosis in melanoma and leukemia mouse models in vivo [34,36]. Apart from its apoptosis-inducing ability, thioridazine has been demonstrated to play various functions such as DNA damaging triggering, anti-angiogenesis, cell cycle arrest via the PI3K or P53 pathway, and anti-calmodulin activity [34,37,38]. These functions are strongly correlated with cancer cell proliferation and survival. Increased sensitization was reported following combination treatment with loratadine and thioridazine in a gastric cancer model [39]. Clinical trials of thioridazine in various cancer types have been reported. A recent example is a Phase I study of a combination of thioridazine and cytarabine in patients with acute myeloid leukemia [40]. This study showed promising results and further clinical trials could support thioridazine combination treatments.

Thioridazine targets various drug-resistant cancer cell types, including MDR cancers and CSCs [34,41,42,43]. In particular, thioridazine has a sensitizing effect on various organ-derived CSCs, such as leukemia, breast, lung, colon, and glioblastoma cells. In addition, it increases sensitization in advanced and chemoresistant cancer types, such as ovarian cancer and TNBC, through PI3K/AKT inhibition [43]. Thioridazine specifically sensitizes drug-resistant oral cancer cells more than sensitive parent oral cancer cell lines [23,24,34]. The study showed that thioridazine induced G2 arrest and apoptosis in P-gp-overexpressing drug-resistant KBV20C cells compared to sensitive parent KB cells, suggesting that single treatment of thioridazine can be a targeting therapy for P-gp overexpressing drug-resistant cancers. Combination treatment with the chemotherapeutic drugs carboplatin and thioridazine synergistically increased the cytotoxicity of breast cancer stem cells in metastatic TNBC, which has highly drug-resistant phenotypes [43]. This study used a slightly higher dose of thioridazine (20 μM) in combination therapy to sensitize TNBC.

Through a large screening of libraries of known compounds, a previous study identified thioridazine as a selective target of CSCs [41]. In this study, the anti-malarial drug mefloquine was also found to selectively sensitize CSCs. Both drugs at low doses showed P-gp-inhibitory activity, and their co-treatment sensitized antimitotic drug-treated P-gp-overexpressing drug-resistant cancer cells [24,25]. A low-dose of ˂5 μM thioridazine can be used for combination treatments, suggesting that thioridazine mainly plays a role in inhibiting P-gp activity and increasing the accumulation of co-administered antimitotic drugs in P-gp-overexpressing drug-resistant cancer cells. Another study showed that co-treatment with thioridazine restored sensitization of doxorubicin-resistant mouse leukemia cells [44]. The co-delivery of thioridazine and doxorubicin was developed using polymeric micelles and showed better sensitization effects in CSCs [45]. Among the 15 bipolar drugs, low-dose thioridazine could be one of the top five drugs for sensitizing antimitotic drug-treated P-gp-overexpressing drug-resistant cancer cells [21,23].

Thioridazine seems to be the final repositioned drug considered for cancer therapy. Determining which type or what individuals are most effectively sensitized to low-dose thioridazine remains an issue. Therefore, personalized treatments with thioridazine should be studied for further application in patients with cancer. Because thioridazine has been primarily studied in resistant cancer cells, its application as a first-line combination treatment may be closer.

### 2.4. Trifluoperazine

The anticancer activity of trifluoperazine has been extensively studied in various cancer models [14]. Trifluoperazine single-dose treatments have shown sensitization effects on different CSCs and resistant cancer cells. One study showed that trifluoperazine selectively increased apoptosis in ABCB1-overexpressing drug-resistant cancer cells via increased oxidative stress [46]. This study demonstrated that sensitization by trifluoperazine results from the inhibition of the ATPase activity of the ABCB1 protein. Trifluoperazine has shown high sensitization effects in resistant cancer cells and CSCs. It inhibits mitochondrial functions to sensitize brain CSCs [47]. In addition, it sensitizes metastatic melanoma cells by disrupting autophagy flux [48].

Combination therapy with trifluoperazine has been reported in various chemotherapeutic-resistant cancers. A previous study indicated that doxorubicin + trifluoperazine increased the sensitization of doxorubicin-resistant glioma cells via nuclear translocation of the FOXO transcriptional factor. Sensitization to this combined therapy was also demonstrated by a reduced tumor size in an in vivo mouse xenograft analysis [49]. Another combined therapy study revealed that cisplatin + trifluoperazine sensitizes cisplatin-resistant bladder urothelial carcinoma through downregulation of the anti-apoptotic protein Bcl-xL [50]. Triple combination treatment with trifluoperazine has also been shown to sensitize pancreatic cancer cells [51]. This study showed that trifluoperazine could be co-administered with two chemotherapeutic drugs, gemcitabine, and paclitaxel, in a pancreatic cancer cell model. 

In the test of 15 bipolar drugs for their sensitization of antimitotic-drug-treated P-gp overexpressing resistant cancer cells, low-dose trifluoperazine showed strong sensitization effects [21]. Although the P-gp inhibitory activity of trifluoperazine was deficient compared to that of the positive control verapamil in this study [21], our preliminary data (results with other P-gp inhibitory assays using increased rhodamine 123 substrate incubation) confirmed that trifluoperazine has P-gp inhibitory activity similar to that of fluphenazine and thioridazine (Figure 2). Therefore, we concluded that low-dose trifluoperazine inhibits P-gp activity in sensitizing antimitotic-drug-treated P-gp overexpressing resistant cancer cells. We believe that these studies may contribute to selecting trifluoperazine as a repositioned drug for use as a first-line combination treatment in targeting resistant cancer populations.

### 2.5. Pimozide

Pimozide has shown anticancer activity in various organ-derived cancer cell lines, including breast, lung, liver, prostate, leukemia, and pancreas [15,52]. Pimozide increased the cytotoxicity of these cancer cells by increasing apoptosis, cell cycle arrest, and anti-proliferation. In addition, pimozide inhibits cell migration and invasion in prostate cancer models by regulating N-cadherin and E-cadherin [52,53]. Pimozide is a well-known JAK/STAT pathway inhibitor that specifically downregulates Stat3 or Stat5 activation [52,54]. In addition, it induces autophagy by reduction of Stat3 activity, which increases apoptosis, tumor survival inhibition, and tumor growth suppression [55,56].

Furthermore, pimozide sensitizes ponatinib-resistant cancer cells via Stat5 inhibition and suppression of CSC growth [57]. As a combination therapy, pimozide can be combined with radiation to increase the sensitization of radiation-treated cancer [52]. Pimozide also synergistically increases cytotoxicity when combined with various anticancer drugs such as dasatinib, temozolomide, or cisplatin [52].

Low-dose or 5 μM pimozide showed P-gp inhibitory activity in resistant cancer cells and sensitized antimitotic-drug-treated resistant cancer cells [21,22]. In this study, antimitotic drug + pimozide increased apoptosis, G2 arrest, and DNA damage compared to a single treatment with either antimitotic drug or pimozide. This study demonstrated that low-dose pimozide had P-gp-inhibitory activity similar to that of the well-known P-gp inhibitor, verapamil. Because low-dose pimozide does not affect growth, G2 arrest, or apoptosis in resistant cancer cells [21,22], the P-gp inhibitory activity of pimozide mainly plays a sensitizing role for combination treatment in P-gp-overexpressing resistant cancer cells. As low-dose pimozide has P-gp-inhibitory activity and sensitizes antimitotic drug-treated resistant cancers, it could be a potential candidate for first-line combination therapies in resistant cancer populations.

### 2.6. Quetiapine

Anticancer activity has been demonstrated in hepatocellular carcinoma following a single treatment with quetiapine [15]. Quetiapine increased sensitization in two animal models of hepatocellular carcinoma that were generated by genetic changes [58]. This sensitization by quetiapine involves caspase-related apoptosis through the downregulation of Erk and NF-kB pathways. Co-treatment with quetiapine synergistically increases the cytotoxicity of doxorubicin, paclitaxel, or 5-fluorouracil-treated breast or colon cancer cells [59]. Quetiapine contributes to the sensitization of drug-resistant cancers. Combined therapy with quetiapine increased the cytotoxicity of temozolomide-resistant glioma cells via the inhibition of the Wnt/β-catenin signaling pathway [60]. Low-dose quetiapine was found to have high sensitization effects in antimitotic-drug-treated P-gp-overexpressing resistant cancer cells when tested with 15 known bipolar drugs [21]. Our preliminary data showed that low-dose quetiapine has similar levels of P-gp-inhibitory activity to the same dose of thioridazine, fluphenazine, and trifluoperazine in P-gp-overexpressing resistant cancers (Figure 2). This suggests that the sensitization mechanism of low-dose quetiapine prevents the efflux of antimitotic drugs from inside the cells. Considering that low-dose quetiapine has high sensitization effects on resistant cancer cells, it is assumed that quetiapine could also be a potential repositioned drug candidate for first-line combination therapies.

### 2.7. Loxapine and Iloperidone

Several studies have reported the anticancer activity of loxapine and iloperidone as repositioned drugs [61,62,63]. A recent test of 15 bipolar drugs showed that low-dose loxapine and iloperidone have sensitization effects on antimitotic drug-treated P-gp-overexpressing resistant cancer cells [21]. Although the sensitization effect was much lower than that of previously described bipolar drugs (aripiprazole, fluphenazine, thioridazine, pimozide, trifluoperazine, and quetiapine), both loxapine and iloperidone showed a higher sensitization effect than the seven other bipolar drugs tested [21]. Interestingly, these two drugs showed deficient levels of P-gp-inhibitory activity in our preliminary data (Figure 2), suggesting that P-gp inhibition is crucial for the high sensitization effects in antimitotic drug-treated P-gp-overexpressing resistant cancer cells. As personalized medicine is needed for various first-line combination treatments, the weak sensitization effect of loxapine and iloperidone could contribute to the selection of the best bipolar drug for first-line combination treatments. It is possible that mechanisms other than P-gp inhibition play a role in sensitizing cells to loxapine and iloperidone. For example, they may contribute to the inhibition of antimitotic drug degradation.

## 3. Conclusions

Recent reports have demonstrated that tumors have heterogeneous populations [1,2,3], including drug-resistant cancer cells. This is the main cause of poor survival rates in tumors. Therefore, it is crucial to prevent the proliferation of drug-resistant cancer cells.

In this review, we hypothesized that targeted therapy for drug-resistant cancer cell populations has the benefit of reducing tumor recurrence, which is highly correlated with increased numbers of resistant cancer cells after chemotherapeutic treatments.

Considering that P-gp overexpression is a crucial characteristic of resistant cancers [6,7,64], it could be a successful method to use as a first-line combination treatment targeting P-gp-overexpressing resistant cancer cells. To conduct faster clinical trials for first-line combination treatments, drug repositioning using FDA-approved drugs may be favorable [11,12,13]. Moreover, P-gp inhibition and low-dose treatments are important when selecting a combination treatment. 

Here, we showed that eight bipolar drugs (aripiprazole, fluphenazine, thioridazine, pimozide, trifluoperazine, quetiapine, loxapine, and iloperidone) that are FDA-approved repositioning drugs exhibited sensitization effects at low doses in P-gp overexpressed drug-resistant cancer cells (Table 1). Aripiprazole showed the highest P-gp inhibitory activity and lowest dose among bipolar drugs (Table 1 and Figure 2), suggesting that the potential best candidate is for first-line combination therapy.

Since various bipolar drugs have been developed and showed different degrees of P-gp inhibitory activity, many combination-treatment options for bipolar drugs can contribute to personalized medicine by efficiently targeting P-gp overexpressing drug-resistant cancer. Since bipolar drugs are not substrates for pumping out via P-gp, they can efficiently accumulate and contribute to increasing the cytotoxicity of anticancer drugs in P-gp overexpressing drug-resistant cancer cells. In terms of the benefits of using bipolar drugs as first-line combination treatments, they can be administered orally. Since they have also been developed for their high ability to penetrate the blood-brain barrier, we assume that they can be easily applied to glioma patients in clinics.

## 4. Future Prospects

In this review, we explain the benefits of bipolar drugs. Various bipolar drugs are available in clinics and are advantageous for application in combination therapies based on personalized medicine. Further investigations into bipolar drugs should be accomplished for suitable choices and faster clinical trials of first-line combination treatments. Strong P-gp inhibition can harm normal and immune cells that attack cancer cells in the microenvironment [9,10]. Considering that some bipolar drugs have weak P-gp inhibitor activity but high sensitization effects with combination treatments, more mechanistic studies are required to apply these drugs. For example, bipolar drugs can have inhibitory activity for cytochrome P450 enzymes, which are anticancer drug-metabolized proteins, to reduce inside toxicity [65,66].

The pharmacokinetics (PK) profile (such as absorption, distribution, metabolism, and excretion) is an essential factor to consider for selecting bipolar drugs in first-line combination treatments since drug-drug interaction by combination treatments can change the PK profile of bipolar drugs and diminish their targeting role in drug-resistant cancer populations. In our studies, we have focused more on lower toxicity (low-dose and FDA-approved drugs) in the body for repositioning bipolar drugs. Future research will focus on checking PK values by testing animal models or searching the literature.

Since these drugs are FDA-approved and have been widely used in clinics, we assumed their PK values would be easily available. We also suggest that animal studies should be performed to assess the PK values for combination treatments. This could stimulate faster clinical trials of low-dose bipolar drugs in first-line combination treatments.

We believe that this review will initiate clinical trials using bipolar drugs as first-line combination treatments targeting P-gp-overexpressing resistant cancer populations.

## Figures and Tables

**Figure 1 ijms-24-08389-f001:**
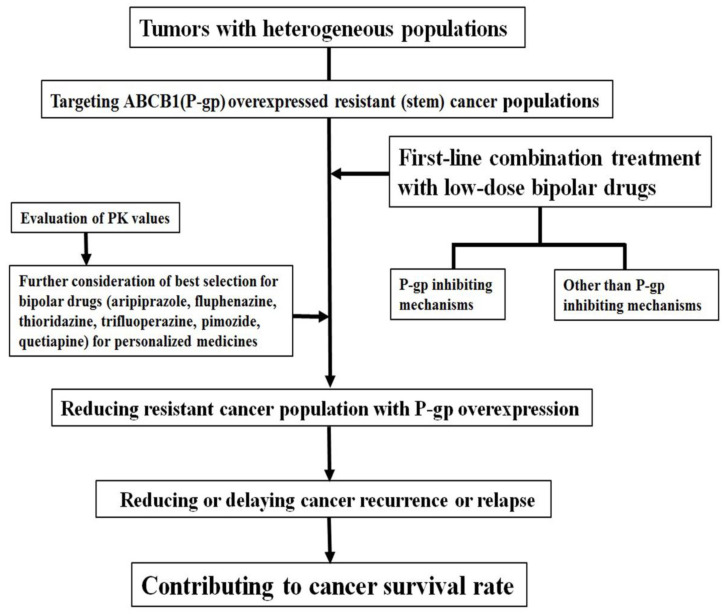
Co-treatment of low-dose bipolar drugs increases sensitization for ABCB1 (P-gp)-overexpressed drug-resistant cancer populations. Survival rates may increase with this first-line combination treatment via removal or delayed proliferation of resistant cancer populations.

**Figure 2 ijms-24-08389-f002:**
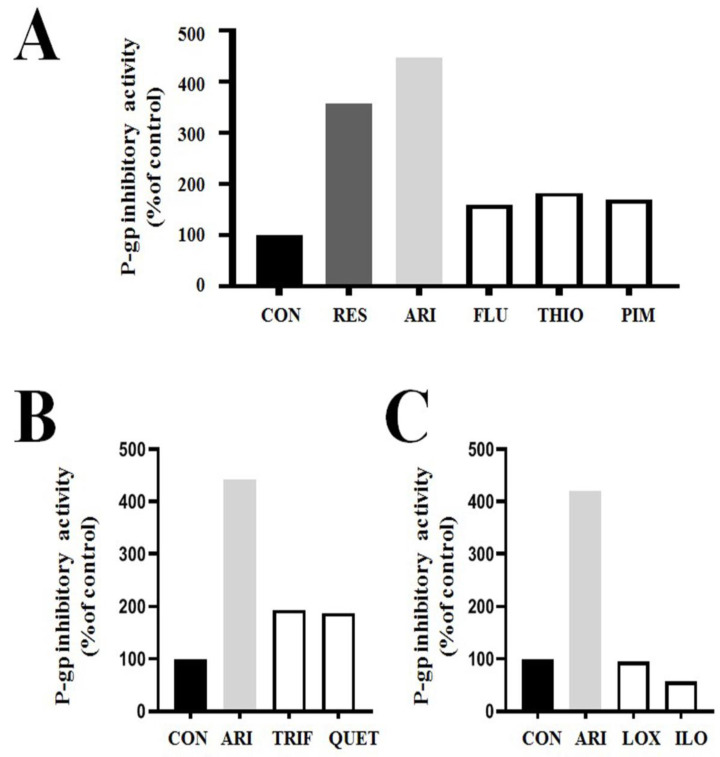
Aripiprazole has the highest P-gp-inhibitory activity among tested bipolar drugs in P-gp overexpressing drug-resistant KBV20C cancer cells (preliminary data). (**A**–**C**) KBV20C cells were treated for 1 h with indicated concentrations of 5 μM reserpine (RES; positive control for well-known P-gp inhibitor), 2.5 μM aripiprazole (ARI), 5 μM fluphenazine (FLU), 5 μM thioridazine (THIO), 5 μM pimozide (PIM), 5 μM trifluoperazine (TRIF), 5 μM quetiapine (QUET), 5 μM loxapine (LOX), 5 μM iloperidone (ILO), or 0.1% DMSO (CON). After adding rhodamine 123 for 3 h staining, all cells were examined using fluorescence-activated cell sorting (FACS) analysis. We confirmed the results by repeated experiments.

**Table 1 ijms-24-08389-t001:** Candidate bipolar drugs for first-line combination treatments.

Bipolar Drugs	Dose for Co-Treatment in Resistant Cancer Cells	P-gp-Inhibitory Activity
Aripiprazole	<2 μM	Extremely High
Fluphenazine	<10 μM	Moderate
Thioridazine	<7 μM	Moderate
Trifluoperazine	<7 μM	Moderate
Pimozide	<5 μM	High
Quetiapine	<7 μM	Moderate
Loxapine	>10 μM	Low
Iloperidone	>10 μM	Low

## Data Availability

Not applicable.

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
