# Peer review of "First-Line Combination Treatment with Low-Dose Bipolar Drugs for ABCB1-Overexpressing Drug-Resistant Cancer Populations"

_ijms, 2023, doi:10.3390/ijms24098389_

Round 1

Reviewer 1 Report

1.       The document could greatly benefit from table summarizing the drugs, mechanism of action, benefits, limitations. Also, if these drugs have any on-going clinical trials for their cancer sensitization capabilities, it would be great to include that as well.

2.       The authors are requested to add more background as to why the application of bipolar drugs would be beneficial in drug resistant cancer. There are other drugs that are currently being used or are reported. What advantage do bipolar drugs offer over those sensitizing agents.

3.       The authors are requested to add a section that summarize challenges from a mechanistic as well as translational perspective that bipolar drugs may face to be leveraged in cancer therapy.

4.       The authors do not mention any challenges with delivery or pharmacokinetics of these drugs. Bipolar drugs may have different PK properties which might hamper their use in cancer therapy. The authors are requested to add that facet to the manuscript.

5.       The manuscript would also benefit from figures, preferably result figures from the report’s authors summarize.

Moderate formatting is needed to ensure correction of grammatical errors.

Author Response

We are grateful to the reviewer for the constructive comments. We appreciate the detailed response and valuable suggestions that helped improve our manuscript.

Following the recommendations of the reviewer, we have made required revisions to the manuscript. If any further changes are required, please do let us know. We appreciate your consideration and look forward to your response.

  1.   The document could greatly benefit from table summarizing the drugs, mechanism of action, benefits, limitations. Also, if these drugs have any on-going clinical trials for their cancer sensitization capabilities, it would be great to include that as well.

Response: We thank the reviewer for the recommendation to summarize bipolar drugs in another table.

Our review focused on low-dose bipolar drugs and combination treatments for targeting P-gp overexpressing drug-resistant cancer populations; therefore, we considered their mechanism of functions as either high or low P-gp inhibitory activity. We found no clinical trials of low-dose bipolar drugs for first-line combination therapy. However, our review aimed to stimulate more clinical trials of low-dose bipolar drugs. Therefore, the presently available information is insufficient to make another table.

As per the reviewer’s recommendation, we described the benefits and limitations of bipolar drugs in first-line combination treatments in the “Conclusion and future prospects” section of the manuscript (please see below).

We would like to use only Table 1, which briefly summarizes eight bipolar drugs and MOA as P-gp inhibitory activity. However, we will consider and collect the information suggested by the reviewer in future studies. Please let us know whether further improvements are needed.

  1. The authors are requested to add more background as to why the application of bipolar drugs would be beneficial in drug resistant cancer. There are other drugs that are currently being used or are reported. What advantage do bipolar drugs offer over those sensitizing agents.

Response: As you mentioned, there are many potential repositioning drugs, besides bipolar drugs, for first-line combination treatments. In our previous studies, we identified various FDA-approved repositioning drugs and published the results. These drugs include TKIs, histamine receptor antagonists (allergy drugs), JAK2 inhibitors, anti-fungal drugs, calcineurin inhibitors, anti-bacterial drugs, and more. Combination treatments utilizing low doses of these drugs increased the cytotoxicity of P-gp overexpressing drug-resistant cancer cells. We assume that these drugs are potential candidates for first-line combination therapies. However, comparing the pros and cons of the drugs with those of bipolar drugs is complicated. In addition, we believe that such comparison may confuse the readers and weaken the advantages of bipolar drugs discussed in this review. Therefore, we would prefer to omit this part in this review. Instead, we may consider writing another review paper on these drugs in the future.

In the present review, as the reviewer suggested, we would like to emphasize only on the advantages of using bipolar drugs in first-line combination treatments. Therefore, we added the following sentences in the “Conclusion” section of the manuscript:

“Since various bipolar drugs have been developed and showed different degrees of P-gp inhibitory activity, many combination-treatment options for bipolar drugs can contribute to personalized medicine to target P-gp overexpressing drug-resistant cancer efficiently.”

“Since bipolar drugs are not substrates for pumping-out via P-gp, they can efficiently accumulate and contribute to increasing cytotoxicity of anticancer drugs in P-gp overexpressing drug-resistant cancer cells.”

“In terms of the benefits of using bipolar drugs as first-line combination treatments, they can be administered orally. Since they have also been developed for the high ability of penetrance for a blood-brain barrier, we assume that they can be easily applied for glioma patients in clinics.”

  1. The authors are requested to add a section that summarize challenges from a mechanistic as well as translational perspective that bipolar drugs may face to be leveraged in cancer therapy.

Response: We divided the “Conclusion and future prospects” section into “3. Conclusions” and “Future perspectives” subsections. We emphasized the advantages of using bipolar drugs in first-line combination treatments, as per the reviewer’s suggestion #2, in the Conclusion section of the manuscript. In addition, we noted that low-dose bipolar drugs could be used. P-gp inhibitory activity can be mechanistic in combination treatments for P-gp overexpressing drug-resistant cancer cells. In the “Future perspective” section of the manuscript, we discussed that this review and further investigations of bipolar drugs could lead to faster clinical applications in first-line combination treatments.

In addition, we included the following sentence:

Here, we showed that eight bipolar drugs (aripiprazole, fluphenazine, thioridazine, pimozide, trifluoperazine, quetiapine, loxapine, and iloperidone) that are FDA-approved repositioning drugs exhibited sensitization-effects by low-dose in P-gp overexpressed drug-resistant cancer cells (Table 1). Aripiprazole showed the highest P-gp inhibitory activity and lowest dose among bipolar drugs (Table 1 and Figure 2), suggesting that the potential best candidate is for first-line combination therapy.

  1. The authors do not mention any challenges with delivery or pharmacokinetics of these drugs. Bipolar drugs may have different PK properties which might hamper their use in cancer therapy. The authors are requested to add that facet to the manuscript.

Response: As you mentioned, we missed an essential part of this review. Our studies have focused on lower toxicity (low-dose and FDA-approved drugs) in the body. As per on your comment, our future research will consider checking PK values by testing animal models or conducting a literature search. Because these drugs are FDA-approved and have been used for a long time in clinical practice, we assumed that their PK properties (absorption, distribution, metabolism, and excretion) could be accessed. Animal studies should be performed to assess their PK properties in combination treatments. This could stimulate faster clinical trials of low-dose bipolar drugs in first-line combination treatments.

We added the following sentences in the “Prospects” section of the revised manuscript:

“Pharmacokinetics (PK) profile (such as absorption, distribution, metabolism, and excretion) is an essential factor to consider for selecting bipolar drugs in first-line combination treatments since drug-drug interaction by combination treatments can change PK profile of bipolar drugs and diminish targeting role in drug-resistant cancer populations. In our studies, we have more focused on lower toxicity (low-dose and FDA-approved drugs) in the body for repositioning bipolar drugs. Future research will focus on checking PK values by testing animal models or searching the literature. Since these drugs are FDA-approved and long-time used drugs in clinics, we assumed their PK values would be easily available. We think animal studies should be performed for assessing the PK values for combination treatment. This could stimulate faster clinical trials in low-dose bipolar drugs in first-line combination treatments.”

Because we realized that PK value is one of the critical parameters for selecting the best bipolar drugs for use in combination treatments, we added “Evaluation of PK value” to Figure 1.

  1. The manuscript would also benefit from figures, preferably result figures from the report’s authors summarize.

Response: As per your suggestion, we included Figure 2, which shows unpublished preliminary data for P-gp inhibitory activity with bipolar drugs. In our previous published studies, P-gp inhibitory activity was measured with a relatively short incubation time (less than 2 h) of rhodamine 123 substrates. We obtained less P-gp inhibitory activity with positive controls (verapamil or tariquidar) with short incubation time of rhodamine 123. We increased the incubation time of rhodamine 123 with over 3 h. With this newly introduced P-gp inhibitory assay method, we could observe detailed degrees of P-gp inhibitory activity. We added the preliminary results to the present review. We assumed that Figure 2 will show the highest P-gp inhibitory activity for aripiprazole. In addition, the figure can help readers understand the P-gp inhibitory activity of various bipolar drugs for Table 1.

Following is the legend of Figure 2: “Aripiprazole has the highest P-gp-inhibitory activity among tested bipolar drugs in P-gp overexpressing drug-resistant KBV20C cancer cells (unpublished data). (A–C) KBV20C cells were treated for 1 h with indicated concentrations of reserpine (RES; positive control for well-known P-gp inhibitor), aripiprazole (ARI), fluphenazine (FLU), thioridazine (THIO), pimozide (PIM), trifluoperazine (TRIP), quetiapine (QUET), loxapine (LOX), iloperidone (ILO), or 0.1% DMSO (CON). After adding rhodamine 123 for 3 h staining, all cells were examined using FACS analysis. We confirmed the results by repeated experiments.”

Reviewer 2 Report

  • Overall, this article is good to publish with minor revisions.
  • This review was nicely designed and describes several bipolar drugs that have recently been shown to sensitize P-gp-overexpressing drug-resistant cancer cells which can be used as potential candidates for first-line combination treatments and repositioned at low doses.  Methods are clearly explained, detail description provided in general and gradually going into specific details.
  • Each section is demonstrated with adequate details.
  • Appropriate references were cited.
  • Abbreviations elaboration in beginning is recommended to remind the reader in the beginning of the article. 
  • Conclusion part can be little more specific with additional detailing with respect to the future directives. 
  • Providing doi's for the references is recommended.
  • Indent change is recommended for figures to maintain consistency throughout the review with the text.

English is 

Author Response

Response: We thank the reviewer for the constructive comments.

: We have made necessary revisions to the manuscript based on these recommendations. If any further changes are required, please do let us know. Thank you for your consideration, and we look forward to your response.

  • Overall, this article is good to publish with minor revisions.
  • This review was nicely designed and describes several bipolar drugs that have recently been shown to sensitize P-gp-overexpressing drug-resistant cancer cells which can be used as potential candidates for first-line combination treatments and repositioned at low doses.  Methods are clearly explained, detail description provided in general and gradually going into specific details.
  • Each section is demonstrated with adequate details.
  • Appropriate references were cited.

Response: We thank the reviewer for the constructive comments. We truly appreciate the detailed response and valuable suggestions that improve our study.

  • Abbreviations elaboration in beginning is recommended to remind the reader in the beginning of the article. 

Response: We thank the reviewer for this suggestion. As you mentioned, we added essential abbreviations in the abstract (right below keywords) and removed the abbreviation part at the end of the manuscript.  

  • Conclusion part can be little more specific with additional detailing with respect to the future directives. 

Response: We thank the reviewer for this suggestion. As you suggested, we realized it would be better to divide the “3. Conclusion” and “4. Future perspectives” sections. We added a “3. Conclusion” section about solid points for using bipolar drugs in first-line combination treatments.

  • Providing doi's for the references is recommended.
    Response: We thank the reviewer for this suggestion. As you mentioned, we made the necessary changes.
  • Indent change is recommended for figures to maintain consistency throughout the review with the text.

Response: We thank the reviewer for this suggestion. As you mentioned, we made the necessary changes.

Round 2

Reviewer 1 Report

The authors have satisfactorily addressed the comments.

Minor editing may be required